# African Elephant Milk Short Saccharide and Metabolite Composition and Their Changes over Lactation

**DOI:** 10.3390/ani13030544

**Published:** 2023-02-03

**Authors:** Gernot Osthoff, Irenie Wiese, Francois Deacon

**Affiliations:** 1Department of Microbiology and Biochemistry, University of the Free State, Bloemfontein 9301, South Africa; 2Department of Animal, Wildlife and Grassland Sciences, University of the Free State, Bloemfontein 9301, South Africa

**Keywords:** elephant, lactation, milk, metabolite, oligosaccharide

## Abstract

**Simple Summary:**

Elephant milk composition differs from all other mammals. The changes of elephant milk composition throughout lactation are also unique. The milk components are synthesized by metabolic processes and leave traces of molecules that provide clues of how these processes are aligned or which of these processes become preferred. Previous research showed that, in elephant milk, a variety of milk sugars are produced during lactation, and the fat content increases over lactation. This was evidenced in the current research by high contents of metabolites of the relevant metabolic pathways. The metabolites also show that some of the pathways may become less or more active with the progression of lactation.

**Abstract:**

Elephant milk composition is unique, as are its changes over lactation. Presented here is the milk non-dedicated metabolite composition of three African elephants. Their lactation times are overlapping and span day one to thirty months. Metabolites were identified and quantified by ^1^H nuclear magnetic resonance spectroscopy. Lactose and short oligosaccharides are a large component of the metabolites, with lacto-N-difucohexaose I as the major oligosaccharide. These were followed by metabolites of lipids, amino acids, and the citric acid cycle. The content of lactose, lacto-N-difucohexaose I, 2′-fucosyllactose, and some unidentified oligosaccharides decrease over lactation, while that of difucosyllactose and other unidentified ones increase. The high content of glutamate, as a glucogenic amino acid, supported the uprated synthesis of saccharides by the milk gland cells. The content of succinate and choline increase over lactation, indicating higher energy expenditure and phospholipid synthesis during later lactation.

## 1. Introduction

The composition of elephant milk is unique. This accounts for African- (*Loxodonta africana*) [1,2] as well as Asian elephants (*Elephas maximus*) [3,4]. It contains high levels of lactose as well as oligosaccharides, and some of the latter are unique to elephants [5,6]. The fat content may reach up to 15% in late lactation, contains between 30 and 60% capric acid [2] and has a low melting point [7]. The protein component lacks α-caseins [8], and β-casein is the major protein [9]. These properties seem to be phylogenetically determined and, although some individual properties seem to be shared with other members of the Atlantogenata, the combined differences are only observed for the Proboscidea [10]. All these components change throughout lactation, with a decrease in saccharides and an increase in fat and proteins, and it has been established that these changes are not affected by seasonal environmental conditions or nutrition [1,2,3].

The metabolome of human breast milk has been studied in some detail [11,12,13]. Less is known of the milk metabolome of animals, and most reports focus on domesticated animals such as the cow (*Bos taurus*) [14,15], goat (*Capra hircus*) [16], and Bactrian camel (*Camelus bactrianus*) [17]. Information on milk metabolomics finds application in dairy herd authenticity [18], animal nutrition [19], and food quality and traceability [20]. Very little information is available for other species [21].

It is complicated to compare research reports on milk metabolomes with each other because the focus of each may differ, including whether targeted or non-targeted metabolites were investigated. Concurrently, the extraction methods may also differ. Some reports focus on the hydrophobic metabolites for which analysis by gas chromatography (GC) is preferred [16,22], while liquid chromatography (LC) is used for the study of the hydrophilic metabolites [17,21]. Mass spectrometry (MS) is normally employed for detection in both these chromatographic analyses. In recent years, ^1^H-nuclear magnetic resonance spectroscopy (^1^H-NMR) has become popular as an analytical method for the characterization, detection, and quantification of metabolites [11,23].

Nevertheless, dedicated comparative studies showed that differences in metabolites occur between species and between groups within species as well as between different stages of lactation. A comparative GC–MS study between goat and cow milk showed that out of forty metabolites, ribose, valine, and glycine were specific to goat milk; talose and malic acid were specific to cow milk [16]. A study by LC–MS and NMR distinguished between milk from Holstein and six other dairy species where 68, 74, 54, 58, 77, and 91 metabolites were respectively observed as significantly different between Holstein (*Bos taurus*) and Jersey (*Bos taurus*), buffalo (*Bubalus bubalis*), yak (*Bos grunniens*), goat, camel, and horse (*Equus caballus*) [18]. The relative concentrations of lactate, acetate, choline, succininate, and pyruvate were significantly higher in Holstein, and the relative concentrations of carnitine, uridine, and pyroglutamate were significantly lower in Holstein than in the other animals. Furthermore, the relative concentrations of caprate in Jersey milk, hydroxyoctadecadienoate and linoleic acid in buffalo milk, 2-(3-carboxy-3-aminopropyl)-L-histidine in yak milk, caprate and indolelactate in goat milk, and caprate in horse milk were significantly different from those in Holstein milk. In summary, this indicated that the glycerophospholipid metabolism as well as valine, leucine, and isoleucine biosynthesis were prominent in the ruminants, Jersey, buffalo, yak, and goat, while the biosynthesis of unsaturated fatty acids was shared in the non-ruminant animals, Bactrian camel, and horse [17,18].

In two species, a change of milk metabolites over lactation was reported. Within a species, an LC–MS study of colostrum and mature milk of donkey (*Equus asinus*) [21] showed that out of two hundred metabolites, seventy were characterized in both the colostrum and mature milk, while fifty-two of the metabolites in the colostrum were significantly different from those in the mature milk. Of these, 8 were downregulated and 44 upregulated. In human breast milk, a total of 159 metabolites were characterized by GC–MS, of which 72 were differentially expressed between colostrum and mature milk [22]. Of these, 17 were upregulated and 55 downregulated in the colostrum and were related to glycine, serine, and threonine metabolism; glyoxylate and dicarboxylate metabolism; alanine, aspartate, and glutamate metabolism; pentose and glucuronate interconversions; and aminoacyl-tRNA biosynthesis.

A major component of human breast milk is oligosaccharides, which play an important role as prebiotics, antiadhesive antimicrobials, and immunomodulators [23]. It has been established that milk oligosaccharide composition differs between species and that some oligosaccharide structures are unique to certain species [24,25]. It has even been reported that the oligosaccharide composition may differ between human ethnicities [26,27] or phenotypes [11,13] and that the content changes over lactation [11,26,27,28]. The amounts of oligosaccharides present in milk as well as their structural properties seem to differ phylogenetically, with the type I chain (Gal(β1-3)GlcNAc) being more prominent in human milk than the type II chain (Gal(β1-4)GlcNAc). The role of the type I chain as a prebiotic may be related to the specific growth of bifidobacteria in the human infant colon [26]. Some species, such as some great apes and the elephant, contain only the type II chain [5,6,26].

As previously mentioned, it was shown that the composition of elephant milk is unique and changes over lactation. It would therefore be interesting to find out whether the uniqueness is extrapolated to the milk metabolome. The results could be important for the nutrition of elephant infants. It would also be the first study of a species whose milk content is characterized by a simultaneous high content of lactose as well as oligosaccharides. Here, we report an initial study of the short oligosaccharides and metabolites related to saccharide and fat synthesis in the milk of three African elephants of overlapping lactation time. Most of the milk samples used here were the same for which the milk composition was reported in a previous study [2].

## 2. Materials and Methods

### 2.1. Animals and Sample Collection

Milk samples were obtained from three healthy and free-roaming African elephants at different lactation stages after a first parturition. All three were approximately 15–17 years old. Shorty and Shan bore male calves and Mussina a female. Elephant Shorty provided milk for the complete lactation, from day one up to 19 months. This elephant lived in Pamuzinda Safari Lodge in the Mashonaland central province, Zimbabwe. In early life, Shorty was used for elephant rides for some years, but this practice was stopped. Interaction with tourists was continued, however, which necessitated daily obedience training of 1–4 h per day depending on the number of tourist interactions. During obedience training, Shorty was fed a few handfuls of 12% grazer game cubes (Reg. no. V19237 Act 36/1947, EPOL Feeds, Durban, South Africa). The rest of the day and night, the elephants roamed freely and fed on natural vegetation. The riding and obedience training in early life made Shorty used to human contact and handling and provided trust to allow milk collection without tranquilization and without causing stress. CITES permit numbers 164211 and 199643 were obtained for importing the milk. Elephants Mussina and Shan roamed in the Adventures with Elephants Reserve, Bela Bela, Limpopo province, South Africa, and provided milk for lactation from months 11–18 and 19–30, respectively. The daily routine was similar to that of Shorty above, but feeding was supplemented with oat hay and bana grass (a hybrid *of Pennisetum americanum* and *P. purpureum*).

Milk was drawn at weekly intervals (Shorty) during the first month and every two weeks (all three elephants) thereafter. While the baby suckled from one teat, milk was collected from the other. This practice was necessary because milk letting is stimulated by suckling [29]. Teats were milked out as far as possible to avoid possible variation of nutrient content during a milk collection session. Milk samples were stored frozen and kept at −23 °C during transportation. Once in the laboratory, they were thawed at 39 °C in a warm water bath with gentle swirling, sub-divided in appropriate volumes for individual analytical procedures, and re-frozen. This was done to keep subsequent thawing and re-freezing steps at a minimum. Due to the complexity and expense, single NMR analyses were carried out but was followed by a duplicate if large deviations from a compositional tendency were noted.

### 2.2. Sample Preparation of Milk for NMR Analysis

NMR analysis was performed according to the method of Erasmus et al. [30], modified for extraction from milk. The milk samples were filtered using Amicon Ultra—2 mL centrifugal units with 10 kDa membrane filters (Merck; Ref UFC201024). Each centrifugal unit was pre-rinsed five times using 2 mL double distilled water at 4500× *g* for 15 min in a Hettich EBA12 centrifuge with 1115 swing-bucket rotor (Hettich AG, Bäch, Switzerland). The rinsing removes trace amounts of glycerol from the membrane filters, which can interfere with NMR signals. One mL of milk was placed in an Eppendorf tube and centrifuged at 12,000× *g* for 5 min in a Hettich EBA12 centrifuge with a Universal 30 F rotor (Hettich AG, Bäch, Switzerland), and 600 μL serum was then filtered as described above. A quantity of 60 μL NMR buffer solution (1.5 M potassium phosphate solution in deuterium oxide with internal standard 580.5 μM trimethylsily1-2,2,3,3-tetradeuteropropionic acid {TSP}, pH 7.4) was added to 540 μL of filtered serum. The sample was mixed by vortex to ensure that it was completely homogenous and then transferred to a 5 mm NMR tube for analysis.

### 2.3. H-NMR Analysis

The prepared samples were subjected to NMR spectroscopy on a Bruker Avance III HD NMR spectrometer (Billerica, MA, USA), equipped with a triple resonance inverse (TXI) ^1^H{^15^N, ^13^C} probe head and x, y, z gradient coils, at 500 M Hz. The ^1^H spectra were acquired as 128 transients in 32 K data points with a spectral width of 6002 Hz and acquisition time of 2.72 s. Receiver gain was set to 64. The sample temperature was maintained at 300 K, and the H_2_O resonance was pre-saturated by single-frequency irradiation during a relaxation delay of 4 s, with a 90° excitation pulse of 8 μs. Shimming of the sample was performed automatically on the deuterium signal. Fourier transform and phase and baseline correction were performed automatically. Software used for NMR processing was Bruker Topspin v3.5 (Bruker BioSpin, Billerica, MA, USA). NMR spectral analysis, peak annotation, and quantification were performed using Bruker AMIX (V3.9.14).

### 2.4. H-NMR Processing

Transformed spectra were corrected for phase and baseline distortions using Topspin v3.5 and then automatically calibrated to TSP as chemical shift reference at 0.00 ppm. The raw data were multiplied by a 0.3 Hz exponential line broadening before applying Fourier transform. Identification of signals was undertaken using Chenomx v9.0 (Chenomx, Edmonton, AB, Canada) or available assignments in the literature. The peaks of the identified metabolites were fitted by a combination of a local baseline and Voigt functions based on the multiplicity of the NMR signal. The missing values were imputed using the k-nearest neighbour (kNN) algorithm [31] (with k = 5). Each metabolite was scaled to its mean value. Quantification was by R software (R Core Team, 2018). For the unknown metabolites and the oligosaccharides that were not positively identified, the number of protons of unassigned signals were imputed equal to 1 and expressed as arbitrary units (a.u.).

### 2.5. Statistical Analysis

Individual scatterplots of time into lactation against individual chemical attributes were constructed for the data of each elephant separately. The data of all animals combined and exponential, linear, logarithmic, and polynomial regressions were alternately fitted to each graph to determine the best fit (with respect to equation and the R^2^ value) between time into lactation and chemical attributes by using the trendline function of Excel 2016 [32].

## 3. Results

The ^1^H-NMR spectra of the three milk extracts at 2, 11.8, and 29.5 months of lactation as examples of early, mid, and late lactation are shown in Figure 1. The signals of the identified metabolites using Chenomx are listed in Table 1. A total of 36 metabolites were distinguished, of which 14 were unknown. The major component of the metabolome consisted of carbohydrates, specifically lactose and short oligosaccharides. Of twelve oligosaccharides, four could not be identified while eight were identified as previously described. Lacto-N-fucohexaose was the major oligosaccharide. Five carboxylic acids were identified, as well as choline and two of its phosphorus-derivatives, and three amino acids. The average and range amounts of the metabolites in mature African elephant milk are listed in Table 2. The milk is considered mature after 12 months of lactation because the content of most components has stabilized after that period of time [2,3].

While the content of some African elephant milk metabolites remains constant over lactation, such as acetate and fumarate, others undergo significant changes over lactation, such as choline, succinate, lactose, and some of the oligosaccharides. This is shown by the differences between the data of the two groups of lactation times in Table 2. Detailed changes are shown in Figure 2, Figure 3, Figure 4 and Figure 5. The changes in the milk of each individual elephant are shown in the figures while the trend line was obtained from the collective data of all three elephants.

The greatest changes over lactation were observed in the saccharide component. In the first 12 months of lactation, the lactose content changed from 2018 to 400 mM, after which it changed to approximately 100 mM after 24 months (Figure 2A; R^2^ = 0.9024). A few high values of approximately 800 mM were noted at 13–19 months. The total identified oligosaccharide content also decreased from 1200 to 400 mM after 30 months (Figure 2B). The R^2^ of 0.642 seems low because closer inspection shows the amounts to vary between 800 to 1300 mM during the first two months, mainly due to the difference in the onset of the decrease in the milk of Shorty at three months and Mussina after 12 months. After 12 months of lactation, the total oligosaccharide content continued to decrease from 800 to 500 mM.

Lacto-N-difucohexaose I is the major oligosaccharide and the main role player in the changes of the total oligosaccharide content. Its content changed from 1100 to 150 mM (Figure 2C; R^2^ = 0.585) with a similar increase from approximately 600 to 1100 mM during the first two months and a later onset of the decrease in the milk of Mussina compared with Shorty.

The content of 3′-sialyllactose occurred at much lower amounts at 84–187 mM and seemed to change from a low 50 mM to a maximum of approximately 170 mM and then returned to a low 80 mM (Figure 2D). The R^2^ of 0.2709 of this tendency is low, but we do want to show it because a constant increase from 50 to 187 mM was observed in the milk of Shorty. The inter-individual variance for this oligosaccharide seems to be high.

The content of 2′-fucosyllactose decreased over lactation from 2.5 mM to an average of 1 mM after 12 months (Figure 2E; R^2^ = 0.5167). The content of 3-fucosyllactose and difucosyllactose increased over lactation (Figure 2F,G), respectively, from approximately 12 to 100 mM (R^2^ = 2636) and 5 to 15 mM (R^2^ = 3882). The R^2^ values are low due to inter-individual differences at the low content.

Four oligosaccharides that could not be identified occurred at less than 100 a.u. Unidentified oligosaccharides 1 and 3 increased, respectively, from approximately 10 to above 40 a.u. (Figure 3A; R^2^ = 0.4837) and 2 to above 65 a.u. (Figure 3B; R^2^ = 0.6067) during the first 12 months, after which their content remained constant. The inter-individual differences of these oligosaccharides were high. The content of unidentified oligosaccharide 4 decreased from 25 to 2 mM during the first three months of lactation and remained constant thereafter (Figure 3C; R^2^ = 0.7581).

The metabolites that did not undergo any changes over lactation (Table 2) are acetate at 3–11 mMol, acetylcarnitine at 0.04–0.49 mM, creatine at 12.00–28.48 mM, formate at 0.69–2.20 mM, fumarate at 0.06–0.15, hippurate at 0.39–7.00 mM, a phenylalanine derivative at 0.44–13.46 mM, and phosphocholine at 3.01–121.19 mM. Caprate occurred at 28.34–340.82 and glutamate at 79.59–457.21 mM, and neither of the latter two changed significantly over lactation; however, they occurred at high concentrations.

The succinate content occurred at 0.44–3.31 mM. During the first ten months of lactation the content was between 0.5 and 1.0 mM and increased to between 2 and 3 mM after 30 months (Figure 4A; R^2^ = 0.7497). Choline occurred at 3.47–20.40 mM and increased from 3.47 mM to approximately 12 mM during the first ten months of lactation and increased further to above 15 mM after 30 months (Figure 4B; R^2^ = 0.5917).

There were 14 unknown components detected, of which numbers 1–11 and 15 occurred at less than 30 a.u. and numbers 12–14 occurred at between 50 and 100 a.u. Detailed discussion will only be devoted to the four that changed over lactation. The amount of unknown 5 decreased from 17 to 4 a.u. during the first 12 months of lactation and increased to approximately 40 a.u. after 30 months (Figure 5A; R^2^ = 0.532) with large inter-individual differences having occurred. The content of unknown 6 increased steadily over lactation from 5 to 39 a.u. (Figure 5B; R^2^ = 0.797). Unknown 13 also decreased during the early months of lactation from 560 to 60 a.u., after which it increased to approximately 300 a.u. (Figure 5C; R^2^ = 0.6138), while unknown 14 increased from 200 to 680 a.u. and then decreased to 100 a.u. (Figure 5D; R^2^ = 0.603).

## 4. Discussion

In this study, lactose was used as a reference to compare the new data with a previous study. A two-sample *t*-test [33] was used to determine any statistical differences between the two lactose analysis methods of African elephant milk by HPLC analysis in the previous study [2] and the ^1^H-NMR of the current work of the same samples presented in Table 2. The significance level was determined as *p* = 0.9357. Although only three African elephants were sampled in the current study, the change in lactose content follows the same logarithmic line of decline (Figure 2A) as was reported for milk obtained from 14 animals [2]. We therefore have confidence that the data of the other saccharides and metabolites reported here should be sufficiently representative for an initial study of the African elephant milk metabolome.

When the current data are compared with the milk metabolome of other species, the first obvious observation is that the African elephant milk metabolome is very different regarding metabolites and their amounts; the comparison provides additional evidence that African elephant milk is unique amongst mammals [10]. The milk is rich in saccharides and very few other metabolites were detected. The short oligosaccharides that could be identified included five neutral saccharides (2′-fucosyllactose, 3-fucosyllactose, difucosyllactose, lacto-N-difucohexaose I, and lacto-N-difucohexaose II) and one acidic saccharide (3′-sialyllactose). Of these six only 3′-sialyllactose was described in the work on oligosaccharides in African elephant colostrum [6]. The four unknown oligosaccharides listed in Table 2 could not be identified as described above nor from the ^1^H-NMR spectra reported previously [6]. Since it was shown that the amounts of lactose and oligosaccharides differ between colostrum and milk [2], it is possible that the amounts of individual oligosaccharides may also differ.

Very few documents have reported on the quantification of milk oligosaccharides; quantification was mostly reported for human breast milk [13,27,33,34]. Amounts in these reports differ as well, mainly due to the different analytical methods that were applied. It was previously determined that African elephant milk contained at least four times more oligosaccharides than human breast milk and that oligosaccharides are the largest saccharide component [5,6,34]. This is confirmed in the current work, where the average content of all twelve oligosaccharides in mature milk is 2.2 times the lactose content (Table 2). The amounts of only four oligosaccharides may be compared with previous research. The 3′-sialyllactose and lacto-N-difucohexaose I content in human breast milk, as determined by 3- methyl-l-phenyl-5-pyrazolone derivatization (PMP), were 11 mM and 225 mM, respectively, compared with 60–180 mM and 90–930 mM in mature African elephant milk. In a different study of human breast milk, the 3′-sialyllactose content was quantified as between 0.09 mM and 0.20 mM by ^1^H-NMR spectral analysis [13]. In the same study, 2′-fucosyllactose and 3-fucosyllactose contents were quantified as 0.2–4.28 mM and 0.02–4.5 mM, respectively, compared with the 0.27–1.90 mM and 40.3–130.3 mM, respectively, in mature African elephant milk. The high content of 3-fucosyllactose compared with the low content of 2′-fucosyllactose indicates a high expression level of fucosyltransferase 3 and a low expression level of fucosyltransferase 2 [12].

Changes in the total oligosaccharide content (Figure 2B) as well as changes in the content of at least five of the seven identified and three of the unknown oligosaccharides were noted. Lacto-N-difucohexaose I was the major contributor to the oligosaccharide composition and it increased during the first month of lactation, reached a peak at 3–10 months, and then steadily decreased thereafter to a sixth of the maximum amount (Figure 2C). Increases of fucosyloligosaccharides during early lactation have been reported in human breast milk [26,34]. The authors ascribed this to a high activity of fucosyltransferase activity, specifically α 2-3/4 fucosyltransferase, in early lactation.

The content of 3′-sialyllactose (Figure 2D) increased from early lactation to stabilize after approximately six months. An increase in this saccharide from colostrum to mature milk has also been reported in donkey milk [22]. In human breast milk, small changes have been reported during 3–6 months of lactation in Lewis-positive secretors (containing the Le gene which determines the presence of α1,4-fucosylated oligosaccharides) [13].

Regarding the other oligosaccharides detected here, 2′-fucosyllactose (Figure 2E) decreased over lactation, reaching more or less stable levels after 12 months. During the early months of lactation, 3-fucosyllactose (Figure 2F) decreased, reaching a minimum at 3–4 months, and then increased to more or less stable levels after 12 months. Difucosyllactose (Figure 2G) increased over lactation, reaching more or less stable levels after 12 months. In human breast milk, the same tendencies were observed, specifically in Lewis-positive secretors.

With regard to the unidentified ones, oligosaccharides 1 and 3 increased over the course of lactation, while oligosaccharide 4 decreased sharply within the first five months (Figure 3). These increases in oligosaccharides observed in African elephant milk are in contrast to the mentioned researchers; however, conflicting results have also been observed by them, which they ascribed to differences in ethnicity [27,33,34]. In newer research, an increase in lacto-N-fucopentaose in colostrum to mature milk had been reported in donkey milk [22].

As mentioned in the introduction, it is difficult to compare the milk metabolite content between species, specifically quantitatively. Nevertheless, a few comparisons are possible. Succinic acid occurs at low amounts relative to the other metabolites in the milk of cows and goats [16], which was also observed in the current study. Its concentration increases throughout lactation (Figure 4A), which indicates an increase in metabolic activity, specifically of the energy-producing pathways such as the citric acid cycle [35]. Caprate occurs at high amounts of 30–340 mM without an obvious trend of change over lactation. The high demand of this fatty acid indicates an uprated fatty acid synthesis by the fatty acid synthase and is required for the synthesis of the milk fat, which may contain up to 60% caprate [2,3]. It occurs in higher amounts in goat milk and was found to distinguish the milk metabolomes from other species [18].

Up to 100 mM phosphocholine was measured, which would implicate an uprated synthesis of phospholipids [36], specifically for the synthesis of membrane material which is required as fat globule membrane [37]. No trend of change over lactation was noted. Phosphocholine plays a key role in the phospholipid synthesis of cell membranes. This process was shown to be upregulated in the mammary gland of cows during lactation [19,36]. Choline was not detected at high amounts, but the levels changed over lactation (Figure 4B), which is perhaps also an indication of elevated synthesis of phospholipids required for the fat globule membrane to cover increasing fat content with progression of lactation [2]. The occurrence of relatively high amounts of capric acid in phospholipids of African elephant milk [2], compared with milk from other species, may also support the elevated synthesis of phospholipids.

The last metabolite of high content is glutamate at 70–370 mM. Glutamate is a glucogenic amino acid that is metabolized to α-ketoglutarate, which enters the citric acid cycle to be changed to oxaloacetate—which is utilized for gluconeogenesis [38,39]. The high glutamate content is therefore directly linked to the high saccharide content in the African elephant milk.

Of the twelve unknown metabolites, only three were detected at high amounts of 400–1000 mmol. The amounts of unknowns 12 and 13 declined over lactation, stabilizing after approximately 12 months of lactation (Figure 4). Unknown 14 increased during the first 10 months of lactation and decreased after 20 months. The trends of change of a few of the minor unknown metabolites are also shown in Figure 4 to indicate that they show similar trends of increase or decrease over lactation as the identified ones. It would be of interest to identify these metabolites by a targeted approach.

To conclude, the uniqueness of African elephant milk, with its high content of saccharides, lipids, and protein together with the changes over lactation, provided an ideal subject to study metabolites that provide reasons for this composition and the changes. It contains mainly saccharides and metabolites, such as the glucogenic glutamate, that support the uprated synthesis of saccharides by the milk gland cells during and throughout lactation. High levels of caprate and phosphocholine indicate the uprated synthesis of lipids, both triacylglycerides and phospholipids, in these cells. Although not present in high concentrations, the contents of succinate and choline increase over lactation, indicating higher energy expenditure and phospholipid synthesis during later lactation.

The milk metabolome presented here contributes information to the uniqueness of African elephant milk and may also aid the improvement of surrogate formulas to feed orphaned elephant calves. It also adds to the detailed information for inter-species comparison of milk composition, specifically to those that have a high content of oligosaccharides. In this aspect, it would be important to determine the same for the milk of other mammals with a high milk oligosaccharide content, such as bears [40]. It would also be important to determine whether the high content of lacto-N-difucohexaose I is a determining factor of the composition of the microbial population of the elephant gut to the same extent as the type I chain oligosaccharides are for the bifidobacteria in the human intestine [26].

## Figures and Tables

**Figure 1 animals-13-00544-f001:**
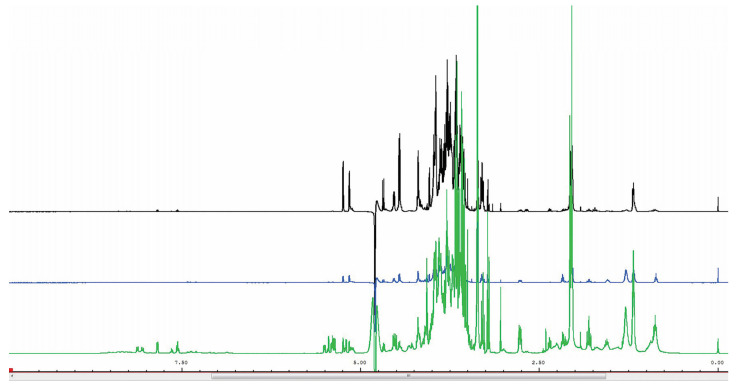
^1^H-NMR spectra of African elephant milk extracts at 2, 11.8, and 29.5 months of lactation (top to bottom).

**Figure 2 animals-13-00544-f002:**
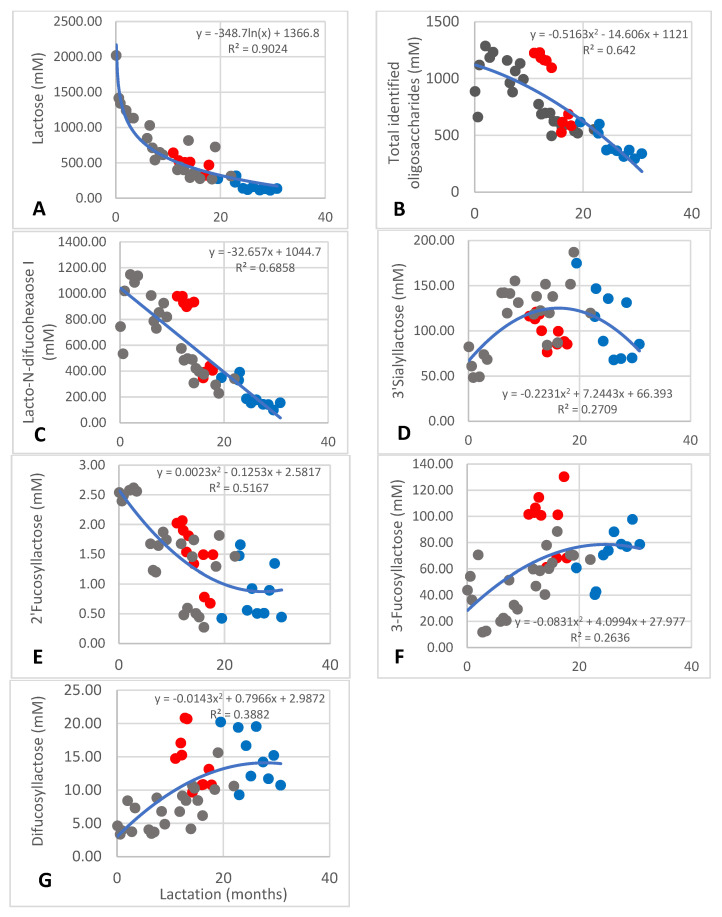
Changes of identified saccharides: lactose (**A**), total identified oligosaccharides (**B**), lacto-N-difucohexaose I (**C**), 3′-sialyllactose (**D**), 2′-fucosyllactose (**E**), and 3-fucosyllactose (**F**) and difucosyllactose (**G**) in milk of African elephants (Shorty ●, Mussina ●, Shan ●). Trend lines, formulas, and R^2^ values were calculated from the combined data of the three elephants.

**Figure 3 animals-13-00544-f003:**
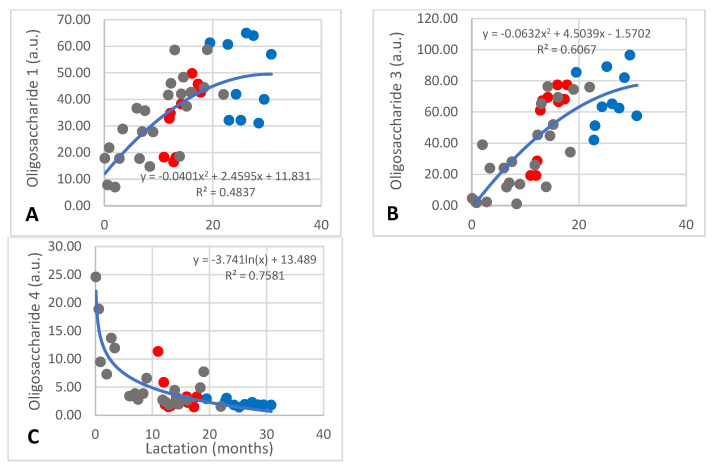
Changes of unidentified oligosaccharides 1, 3, and 4 ((**A**), (**B**), and (**C**), respectively) in milk of African elephants (Shorty ●, Mussina ●, Shan ●). Trend lines, formulas, and R^2^ values were calculated from the combined data of the three elephants.

**Figure 4 animals-13-00544-f004:**
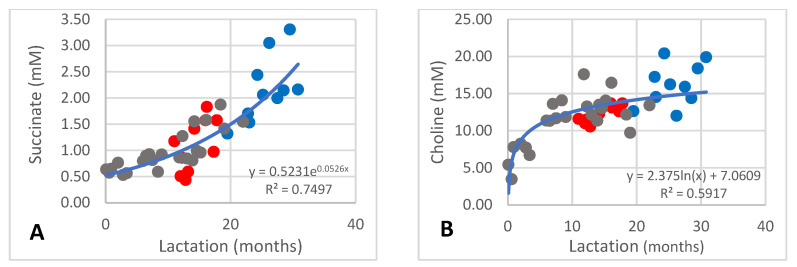
Changes of succinate (**A**) and choline (**B**) in milk of African elephants (Shorty ●, Mussina ●, Shan ●). Trend lines, formulas, and R^2^ values were calculated from the combined data of the three elephants.

**Figure 5 animals-13-00544-f005:**
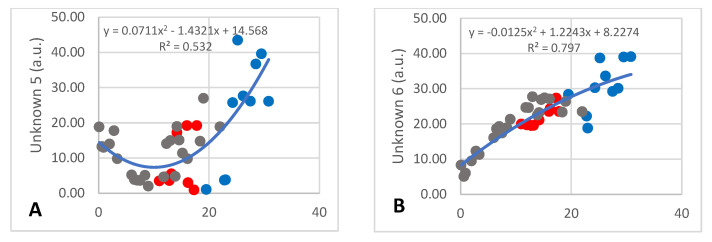
Changes of two minor (**A**,**B**) and two major (**C**,**D**) unidentified metabolites that underwent significant changes over lactation in milk of African elephants (Shorty ●, Mussina ●, Shan ●). Trend lines, formulas, and R^2^ values were calculated from the combined data of the three elephants.

**Table 1 animals-13-00544-t001:** List of the quantified signals and their relative assignment and multiplicity.

Peak for Metabolite or Unknown Compound	Assignment
Formate	8.46 (s)
Unknown 1	8.39 (d)
Unknown 2	8.33 (d)
Unknown 3	8.27 (d)
Unknown 4	8.22 (d)
Unknown 5	8.12 (d)
Unknown 6	8.05 (d)
Hippurate	8.52 (s); 7,84 (s); 7.66 (m); 7.64 (m); 7.63 (m)
Phenylalanine derivate	8.47 (s); 7.44 (s); 7.41 (s); 7.38 (m); 7.36 (m); 7.30 (m); 7.11 (m)
Fumarate	6.52 (s)
Oligosaccharide 1	5.5 (q)
Oligosaccharide 2	5.48 (d)
Difucosyllactose	5.46 (d)
Lacto-N-difucohexaose II	5.43 (d); 2.05 (s)
3-Fucosyllactose	5.44 (d); 5.39 (d)
Oligosaccharide 3	5.36 (q)
2′-Fucosyllactose	5.32 (m); 4.53 (d)
Oligosaccharide 4	5.27 (d)
Lactose	5.24 (d); 4.46 (d)
Lacto-N-difucohexaose I	5.15 (d); 2.07 (s)
Glycerophosphocholine	3.23 (s)
Phosphocholine	3.22 (s)
Choline	3.20 (s)
Acetylcarnitine	3.20 (s)
Unknown 8	3.16 (s)
Unknown 9	3.05 (s)
Unknown 10	3.05 (s)
Creatine	3.04 (s)
3′-Sialyllactose	2.78 (d); 2.75 (d)
Unknown 11	2.44 (s)
Succinate	2.41 (s)
Glutamate	2.37 (s)
Caprate	2.17 (t)
Unknown 12	2.06 (s)
Unknown 13	2.05 (s)
Unknown 14	2.03 (s)
Acetate	1.92 (s)

Underlined indicates the signals used for the following analysis: s = singlet; d = doublet; t = triplet; and m = multiplet.

**Table 2 animals-13-00544-t002:** Metabolite content of mature African elephant milk up to and after 12 months of lactation.

	Up to 12 Months		After 12 Months	
Metabolites	Average ± STD	Range	Average ± STD	Range
Acetate (mM)	7.07 ± 2.33	3.02–10.87	6.34 ± 1.41	4.08–11.06
Acetylcarnitine (mM)	0.15 ± 0.07	0.05–0.31	0.25 ± 0.09	0.04–0.49
Caprate (mM)	117.21 ± 83.84	28.34–315.41	132.64 ± 93.08	32.76–340.82
Choline (mM)	10.23 ± 3.66	3.47–17.60	13.91 ± 2.66	9.72–20.40
Creatine (mM)	16.98 ± 3.08	12.00–23.44	19.90 ± 3.83	15.10–28.48
Formate (mM)	1.48 ± 0.37	1.00–2.30	1.34 ± 0.36	0.69–2.20
Fumarate (mM)	0.09 ± 0.02	0.06–0.13	0.10 ± 0.02	0.06–0.15
Glutamate (mM)	277.55 ± 95.28	125.04–457.21	251.06 ± 69.16	79.59–370.60
Glycerophosphocholine (mM)	24.83 ± 6.32	13.52–38.06	23.37 ± 10.12	6.45–46.28
Hippurate (mM)	3.43 ± 1.90	0.39–7.00	3.80 ± 1.58	1.37–6.56
Lactose (mM)	949.70 ± 437.65	400.7–2018.53	335.10 ± 177.93	110.31–816.01
Phenylalanine derivate (mM)	5.56 ± 3.87	1.26–13.19	4.58 ± 3.79	0.44–13.46
Phosphocholine (mM)	52.90 ± 30.44	3.01–121.19	44.44 ± 35.78	5.99–116.92
Succinate (mM)	0.75 ± 0.19	0.51–1.17	1.56 ± 0.68	0.44–3.31
Unknown 1 (a.u.)	35.07 ± 4.05	25.62–40.98	33.25 ± 6.88	15.47–41.55
Unknown 2 (a.u.)	21.33 ± 2.53	16.76–26.14	21.97 ± 3.73	13.31–28.81
Unknown 3 (a.u.)	35.32 ± 4.41	26.36–42.05	30.48 ± 6.66	14.92–42.40
Unknown 4 (a.u.)	33.35 ± 9.85	18.17–61.27	24.21 ± 7.48	9.88–35.56
Unknown 5 (a.u.)	8.19 ± 5.69	2.03–18.82	16.31 ± 11.93	0.96–43.47
Unknown 6 (a.u.)	15.23 ± 6.03	5.06–24.65	26.46 ± 5.69	18.80–39.08
Unknown 8 (a.u.)	12.02 ± 6.51	0.29–22.88	5.21 ± 3.70	0.49–14.07
Unknown 9 (a.u.)	10.36 ± 2.35	6.04–14.75	16.13 ± 9.35	6.36–41.52
Unknown 10 (a.u.)	10.01 ± 4.94	2.35–18.40	12.58 ± 6.43	2.34–25.76
Unknown 11 (a.u.)	5.59 ± 1.72	3.02–8.76	9.83 ± 2.51	4.52–15.96
Unknown 12 (a.u.)	1017.41 ± 131.31	814.04–14.47	811.58 ± 121.47	562.40–986.88
Unknown 13 (a.u.)	386.50 ± 136.89	158.88–573.39	197.64 ±106.28	54.82–442.13
Unknown 14 (a.u.)	383.43 ± 135.64	170.39–688.69	429.02 ± 183.51	97.17–682.69
Difucosyllactose (mM)	6.77 ± 4.15	3.35–17.07	12.65 ± 4.47	4.17–20.81
2′-Fucosyllactose (mM)	2.02 ± 0.49	1.20–2.62	1.06 ± 0.54	0.27–1.90
3-Fucosyllactose (mM)	44.66 ± 29.17	11.70–102.89	75.15 ± 22.19	40.37–130.30
3′-Sialyllactose (mM)	104.15 ± 36.82	48.27–155.25	112.82 ± 32.37	67.87–187.05
Lacto-N-difucohexaose I (mM)	886.72 ± 188.61	534.12–1147.00	400.77 ± 244.59	98.89–934.54
Lacto-N-difucohexaose II (mM)	7.52 ± 2.91	4.37–13.69	10.00 ± 2.42	5.73–15.12
Total identified oligosaccharides (mM)	1025.10 ± 186.76	659.53–1285.40	651.99 ± 335.71	292.34–1227.70
Oligosaccharide 1	23.63 ± 10.45	7.01–41.67	43.30 ± 13.3	16.48–64.96
Oligosaccharide 2	6.58 ± 9.76	1.13–33.07	7.68 ± 7.14	1.06–30.27
Oligosaccharide 3	15.49 ± 11.53	1.04–39.05	62.88 ± 18.96	11.97–96.54
Oligosaccharide 4	8.65 ± 6.47	2.73–24.58	2.62 ± 1.32	1.42–7.73
Total unidentified oligosaccharides (a.u.)	49.84 ± 16.82	24.26–76.16	199.73 ± 17.88	86.78–150.75

## Data Availability

Raw data were generated at the Centre for Human Metabolomics, North-West University, South Africa. Derived data supporting the findings of this study are available from the corresponding author IDP on request.

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
