# Peer review of "African Elephant Milk Short Saccharide and Metabolite Composition and Their Changes over Lactation"

_animals, 2023, doi:10.3390/ani13030544_

Round 1

Reviewer 1 Report

Simple summary: metabolic pathways are highlighted although pathways were not analyzed per se, consider reformulation

Introduction: Does the genetic background of the elephants influence the oligosaccharide profile (like the secretor status in humans)? Please include references and discussion if possible. The introduction is lacking the current knowledge of the elephant milk metabolites and oligosaccharides and is rather focused on others species.

Materials and methods: Assuming there are obvious challenges related to sample collection, the number of animals and the sample set is not representative, however this is addressed by the authors in the discussion. Please include the age, parity etc. of the elephants if known.

L107–111: Was 2x rinsing tested for efficacy in removing glycerol from the centrifugal units? (personally use 5x)

L113, 144: serum? Do you mean filtrate?

L131: was line-broadening or zero-filling applied?

L132: why alanine signal and not TSP?

L133: please include version no. and company information for Chenomx software

L134–140: which softwares were used for the algorithms, scaling, PCA, KODAMA, …? Please include the information. PCA and KODAMA are missing from the results and discussion.

L137: why ethanol?

L139: why unit variance scaling and not Pareto-scaling?

L153: phophorus

Table 1: the pdf presentation is not optimal for review, so some of the figures are impossible to read. Instead of “Metabolite” maybe prefer “Metabolite/unknown peak” etc., as some of the metabolites may originate from same compounds

Figure 1, Table 1 etc.: Please check the concentration units: mmol … in mL? Should it be mM? Were the metabolites quantified with Chenomx?

Discussion: Please include more discussion of results in comparison to other, although limited number of, studies on elephant milk metabolites and oligosaccharides incl. compared to Asian elephant. As authors mention themselves, the comparison to other mammals is of limited relevance. Are there studies on the nutrition/growth/healt-related requirements of the elephant offspring re milk metabolites/ oligosaccharides that would be relevant to cite in the discussion?

L295–297: please edit sentence for fluency

Reviewer 2 Report

In this work by Osthoff et al., the Authors explore African elephant milk composition, more specifically short saccharides and metabolites, and their changes over lactation. The topic is not much studied and the novelty is thus high. However, there are several points in the manuscript where modifications are needed, for example to clarify the importance/function of each milk component investigated, the detailed explanation of the results and in-depth discussion and implications of the results. Also, the Methodology section should clarify the characteristics of the study animals, explain the living conditions of the animals and not only to cite the previous studies as this should be an independent study. Despite of these shortcomings, I believe the Authors can improve the study further. The longitudinal data is impressive as it spans 30 months, however the sample size of only three mothers is low, but considering the unique nature and rarity of available data, the study gives first very important insights on milk metabolite composition in African elephant milk. 

General comments:

-The Introduction should provide at least some information about the investigated milk components, their functions and why they are important and should be studied? As far as I have understood milk metabolites can however include nutritional components, hormones, pre-biotics, growth factors etc. with multiple different functions, so it should be very clear in the Introduction where the focus is. The only exception here are the oligosaccharides, whose functions are listed in lines 83-88. Also, in the Discussion you should then come back to these things to clarify the implications of the study. I believe these kind of milk studies can provide important insights on elephant conservation related topics too as many of the calves die from lack of milk or potentially poor quality milk.

-Methods: More information about the study animals must be provided and not only cite the previous study. What were the living conditions like and what does free-roaming mean (wild/semi-captive animals?)? How old were the mothers, do you know anything about the health status of the mother? Also, more information about the calf characteristics need to be added, e.g. sex, birth-order and is there allomothers present (do they also allolactate the calf)? I understand it’s not possible to investigate how these characteristics associate with milk metabolite composition (with so low sample size) but these things should be clarified to get an idea how similar/dissimilar the mothers or the conditions were to estimate how much individual and environmental variation there could be in the components. Please elaborate the text and/or add a table of the animals investigated.

-The main results have not been explained in the Results section and they should be elaborated to give a more detailed idea for the Reader what has been found in the study. Now the results section only cites the Figures 1-4 for the main results with changes in the composition over lactation.

-The Discussion lacks detailed discussion of the results and the implications. In the beginning the Authors should give a summary of the main results and then compare the results to previous studies (if not in elephants available, then in other animals). Also, there is no in-depth discussion about the implications, which is very important to advice further studies and to pinpoint why it is important to know about elephant milk composition more in detail (not just to provide comparisons to other animals/humans).

More specific comments:

-lines 14-15: “leave traces of molecules that provide clues of how these processes are aligned or which of these processes become preferred.” Please clarify what is meant by the last part of the sentence.

-line 87: “races”. Should you say ethnicities?

-End of Introduction. Please clarify the study settings and aims, which components where investigated, samples sizes, and hypotheses how these are likely to change with lactation time, if possible.

-line 97-103. More information about the animals and place of study etc. (see above)

 -lines 142-146:  Did you run any statistical tests if the change with lactation time was significant and if so in which components? Also, how was the best fit (exponential/linear/log./polyn.) estimated?

-Results, lines 148-160: All statistics are lacking from the results section and should be added.

-Table 1 (lines 148-149): Should the signal part be moved to supplementary or is it essential that it’s included in the main result section?

-Table 2 (lines 154-156): Why the table 2 only includes samples collected after 12 months of lactation and not all samples that were investigated in this study? If you want to group all samples after 12 months as mature milk, could you have another column grouping samples before 12 months and showing their averages and ranges. This way you would have all data used in the manuscript in the same table.

-Table 1: layout doesn’t work, values go on top of each other in the pdf-file

-Figure 2 legend: Changes of unidentified oligosaccharides…

-line 248: Please explain what is Lewis positive secretors. Also, in humans genetics play a major role in oligosaccharide composition, is it the same for elephants? There are also many other factors that may impact the oligosaccharide composition (e.g. nutrition, season, health status of the mother, infant birth-order/sex) and also potentially other components in milk, do you think similar things could have an impact on the composition of elephant milk?

Round 2

Reviewer 1 Report

General comments: The authors have improved the manuscript after the first round of revision and hopefully spotted and corrected all errors and mistakes in the method description and result presentation etc.  

Since the elephant milk is so unique and not much research has been done, the authors are highly encouraged to include representative NMR spectra of the milk samples (e.g. as supplementary).

Simple summary: Please refrain in key results of this study, e.g. fat content and pathways were not studied here (see the author guidelines).

Materials and methods: Were the metabolites quantified with Chenomx Profiler? How are the unknown metabolites quantified reliably if it is not known how many protons each peak represents? Could you please still clarify this?

L159–160: please use space between the number and the unit

L387–389: conclusions are not made based on this data (lipids and proteins not analysed) and the sentence is very vague

L393: which cells?

Author Response

see file

Reviewer 2 Report

The Authors have carefully addressed all the issues I raised and the manuscript has improved greatly in all sections. Especially the changes to the Methods (study animals etc.), Table 2 and Results section improved the message and quality of paper greatly. Also, the clarifications on the Introduction and Discussion were important. I have only two small comments:

-line 358: Reference is missing

-line 380: Should the sentence say: “Of the fifteen unknown metabolites…”? In table 2 there are listed 15 unknown metabolites.

Author Response

see file
